# Antibody Cross-Reactivity in Auto-Immune Diseases

**DOI:** 10.3390/ijms241713609

**Published:** 2023-09-02

**Authors:** Nicole Hartwig Trier, Gunnar Houen

**Affiliations:** 1Department of Neurology, Rigshospitalet Glostrup, Valdemar Hansens Vej 1-23, 2600 Glostrup, Denmark; 2Department of Biochemistry and Molecular Biology, University of Southern Denmark, Campusvej 55, 5230 Odense M, Denmark

**Keywords:** autoantibody, autoimmunity, cross-reactivity, epitopes, infections, molecular mimicry, virus

## Abstract

Autoimmunity is defined by the presence of antibodies and/or T cells directed against self-components. Although of unknown etiology, autoimmunity commonly is associated with environmental factors such as infections, which have been reported to increase the risk of developing autoimmune diseases. Occasionally, similarities between infectious non-self and self-tissue antigens may contribute to immunological cross-reactivity in autoimmune diseases. These reactions may be interpreted as molecular mimicry, which describes cross-reactivity between foreign pathogens and self-antigens that have been reported to cause tissue damage and to contribute to the development of autoimmunity. By focusing on the nature of antibodies, cross-reactivity in general, and antibody–antigen interactions, this review aims to characterize the nature of potential cross-reactive immune reactions between infectious non-self and self-tissue antigens which may be associated with autoimmunity but may not actually be the cause of disease onset.

## 1. Introduction

Throughout life, humans are at risk of infections, which most often are defeated by the innate immune system. In some cases, when pathogens have developed ways of evading innate immunity or when the immune system is suppressed, more prolonged infections may develop. As a consequence, the adaptive immune system is activated, which results in the production of pathogen-specific T cells and antigen-specific antibodies [1].

In principle, the immune system is capable of responding to any foreign pathogen-associated molecule while remaining tolerant to self-tissues. However, in some cases the immune system fails to differentiate between infectious non-self-antigens and self-tissue, resulting in the generation of immune responses to self-components, also referred to as autoimmunity [2,3].

The development of autoimmune diseases is a complicated process involving several mechanisms [4,5,6]. The mechanism molecular mimicry has been suggested to be associated with the development of autoimmunity, where infectious pathogenic agents activate autoreactive B and T cells in immune-compromised individuals, resulting in an immune response to non-self-antigens, which cross-react with self-components of a similar structure or composition [7,8]. However, the mechanisms associated with molecular mimicry may actually resemble immunological cross-reactivity without necessarily functioning as a main contributor to the development of autoimmunity.

This review provides an overview of immunological cross-reactivity associated with autoimmune diseases, which may contribute to improving our current understanding of immunological cross-reactivity and the development of autoimmunity.

## 2. Autoimmune Diseases

Autoimmune diseases are a heterogeneous group of chronic diseases, affecting approximately 5–10% of the world’s population, with an increase in the Westernized countries [2,9]. More than 80 autoimmune diseases have been identified, including well-known diseases such as type 1 diabetes (T1D), rheumatoid arthritis (RA), multiple sclerosis (MS) and systemic lupus erythematosus (SLE) [10,11,12,13,14].

Autoimmune disorders can be divided into two subclasses: systemic and organ-specific [15]. In organ-specific autoimmune diseases, immune responses are generated against autoantigens limited to a specific organ, resulting in local damage [10,11,16,17,18], whereas in systemic autoimmune diseases, the immune response is directed against ubiquitously distributed autoantigens, resulting in widespread damage [12,13,14,15,19,20].

Differentiation between individual autoimmune diseases can be complicated as autoimmune diseases tend to overlap [21,22,23,24]. Approximately 25% of individuals with an autoimmune disease are prone to develop additional autoimmune diseases, referred to as polyautoimmunity [22,25], which is associated with multiple symptoms, significantly complicating diagnosis and correct treatment [21,25].

## 3. Factors and Mechanisms Associated with Autoimmune Diseases

Defective self-tolerance was originally invoked as an explanation for the occurrence of diseases involving immune reactions to self-components [26,27]. Currently, a common theory is that both genetic dysfunction and environmental factors contribute to the development of autoimmunity (Figure 1) [4,5,17,22,28]. Here, genetic dysfunction relates to predisposition or genetic susceptibility, resulting in that the immune system becomes dysregulated and provides conditions for pathological damage, while environmental factors constitute triggers that make autoimmune diseases clinically apparent [17,28,29].

Genetic dysfunction associated with T cell receptors (TCRs), immunoglobulins (Igs) and major histocompatibility complexes (MHCs), are most commonly linked to autoimmune diseases, whereas reported environmental factors among others include chemicals, drugs, vitamins, smoking and infections [4,5,17,22,28]. Although both genetic as well as environmental risk factors are associated with disease development, it has been estimated that the various risk factors do not contribute equally to the development of autoimmunity, often favoring environmental factors as the main triggers [28].

Infectious agents have emerged as key environmental factors contributing to autoimmunity [17,29,30,31]. In some cases, the pathology has even been considered a “post-infectious” autoimmune disease [32,33]. In particular, Epstein–Barr virus (EBV) has been reported to be associated with various autoimmune diseases, indicating that this virus constitutes a major environmental factor associated with autoimmunity [17,28,34,35,36]. EBV infections are usually obtained during childhood, whereas EBV infections obtained during adolescence may result in mononucleosis [37]. When infected, EBV stays in the host for life and shuttles between an active (lytic) and inactive state (latent) [17,29,35]. Upon EBV reactivation, the immune system may fight off EBV; however, in immune-compromised individuals, the reactivation of EBV is a potential trigger for the initiation of autoimmunity or relapses/flares in EBV-associated autoimmune diseases [30,35,36].

Autoimmune diseases are commonly characterized by the presence of antibodies and/or autoreactive T cells recognizing self-antigens, which ultimately may contribute to tissue damage [7,8,38,39,40,41,42,43,44,45,46,47,48,49]. Tissue damage mainly results from a direct attack on cells carrying the specific self-component, caused by the immune complex formation of local inflammation, where mechanisms such as bystander activation, epitope spreading, and molecular mimicry, have been proposed to play crucial roles [31,50,51,52,53] (Table 1).

Antibodies associated with autoimmune diseases are not necessarily pathogenic; some autoantibodies are reported to be pathogenic and contribute actively to the disease course, whereas others have no pathogenic function and merely function as biomarkers, primarily making them of significant value in the clinical setting [54,55,56,57,58,59,60,61,62,63]. In addition to this, autoantibodies have even occasionally been reported to play a protective role, as some patients with autoimmune diseases have a lower cancer risk [64]. Moreover, it has been noted that decreased levels of self-reactive antibodies may correlate with increased susceptibility to the development of autoimmunity [63,65,66,67], leading to the suggestion that autoantibodies may occasionally have a protective function as well [63,64]. Collectively, the various functions of autoantibodies may point to the suggestion that they do not necessarily contribute to a progressive disease course.

**Table 1 ijms-24-13609-t001:** Proposed mechanisms associated with tissue damage in autoimmune diseases.

Mechanisms	Reference
Bystander activation: implies activation of cells without antigen recognition. During an immune response to non-self pathogens, bystander activation of autoreactive T cells via inflammatory markers, e.g., cytokines, can trigger autoimmunity.	[31,68]
Defective self-tolerance: self-tolerance is maintained through central and peripheral tolerance mechanisms, which entails neutralization of autoreactive cells in the bone marrow/ thymus or in the periphery, corresponding to central and peripheral tolerance, respectively. Breach of these leads to autoimmunity.	[50,51]
Epitope spreading: implies diversification of epitope specificity from the initial dominant epitope-specific immune response to subdominant and eventually cryptic epitopes of the same protein (intramolecular spreading) or other proteins (intermolecular spreading).	[52,53]
Molecular mimicry: implies cross-reactivity between infectious pathogenic antigens and self-antigens, which favor activation of autoreactive T or B cells.	[7,8]

In contrast to autoantibodies, which are produced and secreted upon B cell stimulation and directed to specific targets, T cells involved in immune responses specific for pathogen-derived proteins recognize antigen-derived peptides in complex with membrane-bound MHC I (cytotoxic T cells) or MHC II (T helper cells) [69,70,71,72]. In both cases, the peptides are bound in extended (linear) conformations to a groove formed by two α-helices in the distal extracellular part of the MHC molecules [70,73,74]. In the absence of infection, MHC molecules present self-antigen-derived peptides and this is also the basis for T cell “education” in the thymus, where in-active or self-reactive T cells are eliminated [50]. This process, called central tolerance, involves promiscuous generation and presentation of self-peptides on MHC molecules and assures that T cells expressing TCRs with too low affinity or too high affinity for “self” (MHC-peptide) are eliminated by the processes of positive and negative selection, respectively [50,69,74,75]. TCRs are specific for certain peptides in complex with the MHC molecules, but since the generation of self-peptides in the thymus is promiscuous, and as MHC molecules are rather “promiscuous”, and as TCR interaction with MHC-peptide complexes do not involve all the amino acid side chains of a peptide, there is theoretically a potential for the cross-reaction of pathogen-induced TCRs with some self-peptides, which in principle may result in autoimmune reactions [51].

Recent findings indicate that the potential for cross-reactivity of TCRs is substantial [72,76], in line with the seemingly promiscuous nature of TCR reactivity and T cell selection. This may indicate an important role of co-stimulation/co-inhibition, depending on activation of the innate immune system in preventing auto-immune reactions [77,78]. This is consistent with the current shift in focus from T cells to B cells in the understanding and treatment of autoimmune diseases [79,80].

TCR cross-reactivity and antibody cross-reactivity have been reported to be associated with molecular mimicry, where B and T cells become activated upon the recognition of infectious pathogens (non-self), resulting in immune responses which may cross-react with self-tissue antigens [7,8,15] (Table 1, Figure 2).

The term “molecular mimicry” was originally used to describe the existence of “similar” antigens expressed by infectious agents and an infected host, which may facilitate the ability of microbes to avoid the host’s immune response [81]. One of the first studies describing molecular mimicry was in a patient with rheumatic heart fever, where cross-reactivity between *Streptococcus pyogenes* M protein and the human heart muscle cell protein myosin was reported [82,83].

Cross-reactivity associated with molecular mimicry can be divided into subgroups: structural homology and sequence homology (Table 2) [7,41,84,85,86,87,88,89,90,91]. Sequence homology refers to % identified at the amino acid level, primarily focusing on continuous epitopes, which often present with similar structures (Table 2), whereas structural homology is referred to as similarity at the complete structural protein level, alternatively tertiary and secondary structural levels, and presents with a sequence identity of up to 100% [41].

Structural homology related to antibody reactivity may apply to autoimmune paraneoplastic diseases, for example, the anti-Hu syndrome, where small cell lung tumors for unknown reasons express HuD, a protein expressed normally only in neurons [92,93]. The HuD-specific immune response reacts to the protein expressed in the lungs and the neurons, causing various neurologic symptoms [92,93]. In addition to this, a single example has been reported, where a human protein has been “hijacked” by a virus and presented as an antigen, as cytomegalovirus has been reported to acquire and incorporate CD13 into its viral envelope [90]. This resulted in an immune response to CD13-positive cells such as mononuclear cells, smooth muscle cells, and fibroblasts, associated with graft-versus-host disease [90]. Similarly, structural homology has been reported between human carbonic anhydrase II and α-carbonic anhydrase of *Helicobacter pylori* (*H. pylori*), which led to the suggestion that *H. pylori* infection may trigger autoimmune pancreatitis [41]. However, in general, only a limited number of examples of structural homology have been presented.

Although the presence of cross-reactivity between non-self and self-antigens is a critical criterium in molecular mimicry, other supporting criteria have been proposed [8,91]. These include the establishment of an epidemiological link between the exposure of an infectious agent and onset of autoimmunity, as well as the reproducibility of autoimmunity in an animal model, following sensitization with selected antigenic epitopes upon infection [8,94]. Although proposed, these criteria can be challenging to demonstrate, especially due to limitations of human genetic studies [8]. Based on this, it can be difficult to determine whether the examples presented in the literature describe molecular mimicry or cross-reactivity between infectious non-self-antigens and host tissue antigens without causing tissue damage.

Nevertheless, given the nature of cross-reactivity at the molecular level, the interactions described between an antigen and an antibody/TCR associated with molecular mimicry are expected to follow the same “rules” in terms of cross-reactive antibody/TCR-antigen interactions, ultimately representing two sides of the same coin.

## 4. Antibody Specificity and Cross-Reactivity

Independent of the nature of the antibodies associated with autoimmune diseases, whether being a non-self-infectious cross-reactive antibody or an autoantibody produced in response to self-antigens, all of the antibodies interact with target antigenic epitopes through their paratopes located in the complementarity-determining regions [95,96,97,98]. The interaction between antibodies and antigens is a reversible reaction, based on the summation of several noncovalent interactions [95,98,99,100,101,102]. Antibodies are most often very specific for their targets, primarily interacting with a single target, thus antibody specificity has commonly been described as a measure of the degree to which the immune system differentiates between different antigens [84,98,103,104,105,106].

Occasionally, cross-reactivity occurs where an antibody or a TCR specific for one target recognizes another target [106,107,108,109,110,111,112,113,114,115,116]. Based on this, cross-reactivity can be defined as a measure of the extent to which different antigens appear “similar” to the immune system [106]. Given that a limited number of amino acids found in antigenic epitopes most often contribute to obtaining a sufficient binding energy, cross-reactivity declines rapidly and nonlinearly with the number of amino acid substitutions in the target epitope [106].

The study of antibody reactivity to antigenic epitopes using alanine- and functionality-substituted peptides has provided important knowledge about antibody antibody–antigen interactions and cross-reactivity [84,104,107,108,117,118,119]. These findings have contributed to categorizing epitope–paratope interactions; those that depend on the majority of the amino acid side chains of the epitope for antibody binding; those that depend on a combination of amino acid side chains and backbone in the epitope for antibody binding; and those that mainly depend on a limited number of amino acid side chains in combination with the backbone in the epitope for a stable antibody–antigen interaction [84,104,107,108,117,118,119,120,121]. In most cases, the substitution of critical hot spot residues in the epitope is not tolerated without interfering significantly with antibody binding; not even amino acids with similar side chain functionalities are tolerated in central positions [84,104,107,110,118,119]. Using this approach, antibody cross-reactivity can be determined as the degree of substitutions, which is tolerated in a given antibody–antigen complex, which appear to be more common for antibodies recognizing epitope backbone or a combination of epitope backbone and amino acid side chains [110,117].

The information presented here may not only apply to cross-reactive antibodies but to self-reactive antibodies in general. Thus, knowledge about the nature of antibodies is essential to understand the principle of antibody cross-reactivity and the association of antibodies to autoimmune diseases, independent of whether they are associated with autoimmune diseases as a result of cross-reactivity between infectious non-self-antigens and self-antigens or molecular mimicry.

## 5. Cross-Reactivity in Autoimmune Diseases

Although various types of cross-reactivity have been presented, mainly cross-reactivities between primary amino acids sequences shared between an infectious non-self-antigen and its host have been reported to be associated with autoimmune diseases. The extent of sequence similarity between self- and non-self-antigens may vary significantly; in some cases, the cross-reactive regions have been identified, others only theoretically, while others still remain to be identified [8,110,117].

### 5.1. Rheumatoid Arthritis

RA is a systemic inflammatory disease affecting approximately 1–2% of the population [12]. Although it is a systemic disease, the disease primarily tends to become associated with inflammation located in the joints, as RA is typically characterized by progressive joint damage along with systemic complications [12,122,123].

Genetic as well as environmental factors play crucial roles in the onset of RA [12,124,125,126,127,128,129,130,131]. Genetic factors, especially MHC II alleles carrying a “shared epitope” motif, were originally described to be associated with RA, although other genes such as peptidyl arginine deiminase (PAD) and protein tyrosine phosphatase non-receptor 22 (PTPN22) have been suggested to be associated with RA onset as well [12,128,130,132,133,134]. Moreover, environmental factors such as smoking have been reported to increase the risk of developing RA in predisposed individuals along with various infectious agents, for example *Proteus mirabilis* (*P. mirabilis*), *Porphyromonas gingivalis* (*P. gingivalis*), and EBV [12,85,86,128,130,134,135,136,137].

*P. mirabilis* infections have been suggested to influence chronic inflammation through molecular mimicry, which has been supported by studies illustrating that RA patients have a higher rate of *P. mirabilis* infections [137,138]. These findings are in accordance with serological studies, where antibody cross-reactivity between peptides from hemolysin B and urease C from *P. mirabilis* and human proteins have been reported along with correlations between *P. mirabilis* urease F and disease-specific rheumatoid factors (RF), erythrocyte sedimentation rate, and C-reactive protein in RA patients [139]. Furthermore, cross-reactivity between the human HLA motif EQKRAA and a similar peptide ESRRAL present in bacterial hemolysin has been reported, which has led to the suggestion that molecular mimicry between HLA alleles and *P. mirabillis* may influence the onset of RA [111].

*P. gingivalis* has been reported to be associated with RA as well, although *P. gingivalis* is primarily associated with periodontal disease [85,135,140]. This hypothesis has been supported by associations between RA disease activity and severe periodontitis, as RA patients may have a higher tendency to develop periodontal disease [135,140,141,142]. Similar to human PADs, *P. gingivalis* PAD may citrullinate proteins, a process that may drive the generation of autoantigens in RA, and which represents a common link between RA and periodontal disease [135]. Based on this, *P. gingivalis* has been proposed to be involved in the onset of RA through molecular mimicry as antibodies to a human citrullinated α-enolase have been described to cross-react with a conserved citrullinated sequence found on *P. gingivalis* enolase, which contains an immunodominant epitope important for the detection of disease-specific anti-citrullinated protein antibodies (ACPAs) (DS-Cit-GNPTVE) [85,143]. In relation to this, correlations between antibody levels to bacterial enolase and anti-citrullinated enolase antibodies were reported [85,135,142]. These findings are supported by studies using animal models, illustrating that arthritis was induced by experimental infection with *P. gingivalis* [143,144,145], where *P. gingivalis* was reported to be important in the loss of tolerance to citrullinated proteins in RA [146].

An evident link between EBV and RA development exists [147,148,149,150,151,152,153,154,155,156,157], entailing antibody cross-reactivity as well. Originally, antibodies to a glycine–alanine-rich repeat present in Epstein–Barr nuclear antigen (EBNA)1 have been reported to cross-react with a 62 kD protein present in the synovium of RA patients, leading to the assumption that EBV may contribute to the pathogenesis of RA through cross-reactivity [158,159,160]. Similarly, antibodies to EBNA1 have been reported to be cross-reactive to keratin and denatured collagen [86]. These findings are supported by studies describing that EBV infections in mice may induce erosive arthritis, and the presence of T cell responses in the joints of patients with RA [160,161,162,163]. Another link between RA and EBV is the presence of disease-specific ACPAs, which have been reported to cross-react with citrullinated EBV proteins [164,165]. Here disease-specific ACPAs cross-reacted to citrullinated peptides originating from both EBNA1 and EBNA2 proteins [163,166]. Thorough characterization of the cross-reactive antibodies revealed that these antibodies mainly depend on epitope peptide backbone in combination with a critical Cit-Gly motif for antibody binding [90]. Moreover, the EBNA2 peptide showed strong sequence homology to the filaggrin peptide originally used for ACPA identification in RA [165,167], which may indicate that this peptide contains the epitope closest to the true autoantigen. Based on an evident ACPA cross-reactivity between citrullinated host proteins and EBV-citrullinated peptides, which not only entail citrullinated EBV proteins but also other virus proteins and host proteins, antibody cross-reactivity between ACPAs and citrullinated proteins appears, at first glance, to relate to cross-reactivity between infectious non-self and self-tissue antigens associated with RA, but it does not appear to be the cause of RA; hence, the role of molecular mimicry in the onset of RA is not supported. This hypothesis contradicts the model of *P. gingivalis* as a contributor to the onset of RA through molecular mimicry based on citrullinated epitopes, which has been previously reported [85,135]. In fact, ACPA cross-reactivity observed between citrullinated bacterial enolase and human enolase may reflect the general cross-reactive (and backbone-dependent) nature of ACPAs rather than disease-specific molecular mimicry.

### 5.2. Systemic Lupus Erythematosus

SLE is a chronic systemic autoimmune disease often associated with a characteristic butterfly-shaped rash, covering the cheeks and nose [13]. The disease is characterized by widespread inflammation in the connective tissue, which may affect several organs such as the skin, brain, lungs, kidneys, and blood vessels [13]. Most often, young women rather than men (10:1) suffer from SLE [168,169]. SLE presents with various clinical manifestations and a wide profile of autoantibodies, although primarily antibodies to Ro, La, Smith antigen (Sm), and dsDNA are considered essential, and are also involved in disease diagnosis and progression [170,171,172,173,174,175].

Several genetic as well as environmental risk factors have been reported to influence the dysregulation of cells in the immune system of individuals suffering from SLE, especially in relation to autoantibody production and deposition of immune complexes [176,177,178,179,180,181,182,183,184]. B cell dysregulation is often associated with SLE onset, as B cells mediate the production of autoantibodies and present antigens to T cells [185]. EBV has been suggested to be associated with SLE among others by contributing to disease development through molecular mimicry [8,29,39,112,113,185,186]. In particular, SLE antibody responses to EBV EBNA1 have been widely studied, as humoral responses to EBNA1 have been proposed to generate cross-reactive antibodies in genetically predisposed individuals [187], a finding which is in accordance with studies describing a higher incidence of EBV infection and elevated EBV IgG titers in SLE individuals [186,187,188,189]. Studies using animal models have demonstrated that in vivo expression of antibodies to EBNA-1 elicit the production of antibodies to dsDNA and Sm, possibly through cross-reactivity [112,113,186,190], which is in accordance with identification of regions with sequence similarity between EBNA-1 and Sm (PPPGRRP of EBNA-1 and PPPGMRPP of Sm) [191]. Similarly, regions with sequence similarity have been identified between EBNA-1 and Ro 60, which have been proposed to be associated with SLE-like disease in different animal models [189]. These findings are supported by studies describing antibody reactivity in SLE patients, where Ro 60 antibodies cross-reacted with an EBNA-1-derived peptide, although no sequence similarity between the two peptides was identified, suggesting a possible structural homology [39]. In terms of cross-reactivity to Ro60, a recent study was conducted, assigning the original cross-reactivity between EBNA-1 and Ro60 as a matter of non-specific binding [119]. Hence, the true nature of the cross-reactivity between Ro60 and EBV EBNA1 remains to be further elaborated.

Recently, *Trypanosoma* infections have been described to be associated with SLE [192]. Here, it has been reported that patients following *Trypanosoma* infections may elicit an autoimmune response, leading to the development of SLE-like symptoms [192]. *Trypanosoma cruzi* is a protozoan belonging to the *Reduviidae* family, which was originally described to be associated with Chagas’ disease and is primarily found in North and South America. Recent in silico studies identified 36 autoantigens in SLE, which showed molecular mimicry with *Trypanosoma* antigens at varying levels [192]. Finally, molecular mimicry between SLE and *Leishmania* has been proposed, which, as for *Trypanosoma*, is manifested on a theoretical level, as mimicry primarily was described using theoretical protein sequence search analyses and hence remains to be tested in animal models [193].

Although examples of cross-reactivity and potential molecular mimicry between infectious non-self and self-antigens associated with SLE have been reported on practical and theoretical levels, only limited information about the in-depth characterization of the cross-reactive nature of SLE-associated antibodies has been presented. Based on this, the true contribution of cross-reactive antibodies in SLE cannot be defined, as antibodies reacting to Sm and possibly Ro60 may merely reflect specific antibodies directed to EBNA-1, which cross-react with Sm, rather than antibodies contributing directly to disease development through molecular mimicry. This remains to be further elaborated.

### 5.3. Multiple Sclerosis

MS is a chronic disease of the central nervous system, which is characterized by the inflammation and demyelination of nerve fibers, causing physical and cognitive disabilities [11]. MS is considered to be the most common inflammatory demyelinating autoimmune diseases of the central nervous system (CNS), as more than 2 million individuals are affected worldwide [11]. The disease is most common in women and approximately 85% of individuals are initially diagnosed with relapsing-remitting MS, where relapses are followed by recovery “remitting” although not completely [11].

MS has been considered to be a T-cell-mediated disease, as autoantibodies are only detected in a small proportion of MS patients [193]. The myelin basic protein (MBP), myelin oligodendrocyte glycoprotein (MOG) and the proteolipid protein (PLP) are the main targets of CD4^+^ T cells, although inflammation and damage are believed to be caused mainly by CD8^+^ T cells [194,195]. The etiology of the disease remains to be fully determined, although genetic factors as well as environmental factors are important for disease onset [17,192,196,197,198,199,200]. Especially EBV infections have received attention and are now regarded as mandatory for the development of MS [17,192,196,197,198,199,200]. The risk of developing MS increases 32-fold after EBV infection, which is in accordance with studies describing that elevated EBV EBNA1 antibody titers are typically detected in serum and cerebrospinal fluid (CSF) of MS patients, which have been reported to increase the risk of MS [199,201,202,203,204].

Studies point to that EBV proteins may be associated with MS disease through molecular mimicry [205,206,207]. For example, CD4^+^ T cells from MS patients have been reported to recognize EBV-transformed B cells [208]. Thorough analyses of immune responses in MS patients have described T cells specific for MBP to cross-react with EBNA-1 [208,209,210]. These findings have been supported by structural protein studies, describing structural homology between MBP and EBV peptide fragments presented by human leukocyte antigen (HLA)-DRB1*15:01 [89]. Additionally, autoreactive B cell cross-reactivities have been described between EBNA1 and anoctamin 2 and α-crystallin B chain (CRYAB) [205,206,211,212], which has been supported by the identification of sequences similarity found between EBNA1 and CRYAB and glial cell adhesion protein (GlialCAM) [206,213]. These findings have been supported by studies describing pathogenic antibodies to EBV EBNA1, which were reported to cross-react with GlialCAM found in the CNS in both human cohorts and mouse models, where molecular mimicry was suggested to be facilitated by post-translational modifications of GlialCAM [206]. Finally, antibodies to EBV latent membrane protein 1 (LMP1) have been shown to cross-react with MBP, although this remains to be elaborated [206].

In addition to EBV, several studies have described an association between human herpes virus (HHV)-6 and MS onset, primarily based on increased HHV6-specific antibody titers in MS patients or DNA in MS lesions [213,214,215]. These findings have been confirmed by studies identifying sequence homology between the HHV6 protein U24 and MBP, which has led to the hypothesis that HHV6 may contribute to MS through molecular mimicry [114,115,216].

Although not confirmed, a correlation between the consumption of cow’s milk and the prevalence of MS has been reported, where antibody cross-reactivity between casein and myelin-associated glycoprotein (MAG) has been suggested to result in demyelination in the CNS [116]. In addition, a potential link has been reported between John Cunninham virus (JCV) and MBP from MS patients, which has led to the suggestion that autoreactive T may cells share similar affinity to JCV peptides and MBP, where cross-reactivity occurs [217].

The fact that numerous viruses and both T cell and B cell cross-reactivity may be associated with MS complicates the identification of the precise mechanisms associated with disease development; nevertheless, this significantly reduces the probability that all of the aforementioned viruses contribute to the development of MS through molecular mimicry. Numerous examples of cross-reactive immune reactions, which sparsely describe the type of cross-reactivity, may reflect cross-reactive immune reactions associated with autoimmunity rather than actually being the cause of the disease, especially as none of the examples have been described as being pathogenic. This remains to be elaborated.

### 5.4. Type 1 Diabetes

T1D is most often diagnosed during childhood or in young adults [10]. The disease is characterized by the presence of autoantibodies to the endocrine β cells in the pancreas, which are responsible for the production and storage of insulin [208]. The destruction of β cells results in a significant reduction in insulin production, which over time leads to the development of insulin-dependent diabetes [218,219,220]. Women are more prone to develop T1D and incidence rates vary from 1 to 40 cases per 100,000 individuals per year [220,221].

T1D is believed to be a T-cell-mediated disease, where CD4^+^ and CD8^+^ T cells, B cells, and macrophages infiltrate the pancreatic islets [222,223,224,225]. This theory has been confirmed by findings, describing that the transfer of bone marrow cells from T1D patients to healthy controls predisposes the development of T1D [226].

Genetic and environmental factors have been reported to be involved in the development of T1D [227,228,229,230,231,232]. Infectious agents have been considered to be the main triggers of T1D development, especially as the risk of developing T1D in predisposed children is significantly elevated during cold months, when children especially suffer from viral respiratory infections [230,231,232,233,234,235,236,237,238,239,240]. Especially coxsackievirus B (CVB) has been reported to be associated with T1D development [123,125,241]. Coxsackievirus B (CVB) is one of the major enteroviruses that can be found in T1D patients [242,243]. These assumptions have been supported by the detection of CVB RNA sequences in the peripheral blood of patients at the onset of the disease or during the course of T1D [244]. Moreover, enterovirus infection has been found to be two times more frequent in siblings who later developed T1D than in the control group of nondiabetic siblings [245]. Sequence homology between CVB P2C and GAD65 (PEVKEK) has been suggested to induce T1D through molecular mimicry [246,247], although other studies describing inoculation of non-obese diabetic (NOD) mice with CVB3 illustrate long-term protection from T1D rather than the development of the disease [248,249,250,251]. These findings are in accordance with a recent study, where cross-reactivity between CVB and GAD65 (PEVKEK) associated with T1D was thoroughly characterized [84]. In this study, none of the human sera reacted with peptides containing the homologous PEVKEK sequence from GAD65 and CVB P2C. Furthermore, it was not possible to generate a stable clone, which showed cross-reactivity between the two proteins [84].

Of note, the PEVKEK originating from CVB P2C shows high sequence similarity to GAD67 (PEVKTK), an isoform of GAD65, which is associated with the autoimmune disease stiff-person syndrome [84,252]. Nevertheless, although a high sequence similarity between GAD67 and CVB P2C exists, CVB has not been reported to be associated with the development of stiff-person syndrome, although the two GAD isoforms have high sequence similarity and structural homology.

Finally, rotavirus has been reported to be associated with T1D, as T cell cross-reactivity has been reported between GAD65, tyrosine phosphatase IA-2, and virus proteins [253,254], although this hypothesis has not been confirmed in other studies [255,256].

In general, the nature of cross-reactive immune responses associated with T1D has been sparsely described, which significantly complicates the process of determining whether these are merely associated with the disease or whether they contribute actively to the disease course.

## 6. Discussion

The examples of cross-reactive immune reactions listed in the literature provide interesting knowledge about similarities between infectious non-self and self-tissue antigens associated with autoimmune diseases [76,83,85,86,89,96,106,110,111,112,113,114,115,116]. As presented in the previous paragraphs, cross-reactive immune reactions linked to autoimmunity have been sparsely described in detail, which complicates the interpretation of the results as autoantibodies and T cells may be cross-reactive without being associated with autoimmunity [110,250]. However, based on the characterization of accessible data describing cross-reactive immune reactions and crucial information about the nature of epitopes, antibodies, and antibody/TCR-antigen interactions, the findings support the current hypothesis that cross-reactivity between non-self and self-antigens per se may not result in immune responses towards self-tissues nor contribute to the development of autoimmunity through molecular mimicry [8,248,249].

Many of the mentioned examples of cross-reactivity are often based on partial sequence identity, whereas only a limited number of examples of structural homology exist, complicating the interpretation of such results [8,89,118,119,257]. For example, a study described structural homology between human carbonic anhydrase II and α-carbonic anhydrase of *H. pylori*, which led to the suggestion that *H. pylori* infection may trigger autoimmune pancreatitis [41]. Nevertheless, structural homology is not necessarily sufficient to obtain cross-reactivity, as described in a recent study, where the reactivity of an antibody to the protein calsperin was examined [258]. An immunodominant epitope was determined, which contained a central motif that was identified in another region of the protein. Although both regions contained similar structures and contained a crucial motif for antibody binding, the antibody only reacted to one of the sites [258]. Based on this, it may be concluded that although structural and sequence homology is present, other factors influence antibody cross-reactivity as well.

In contrast to structural homology, sequence homology has most often been reported to be associated with cross-reactivity in autoimmune diseases [85,111,143,191], although studies of sequence homology in cross-reactive regions may point to that cross-reactivity does not necessarily contribute to the development of autoimmunity. Many of the identified sequences describing cross-reactive regions are often composed of five to seven amino acids [89,118,119,257]. Epitopes identified by antibodies are most often longer, typically favoring epitopes of 7 to 15 amino acids, whereas processed epitopes presented to T cell epitopes typically contain 7 to 17 amino acids [259,260]. Hence, based on the length of the cross-reacting epitopes, many epitopes associated with molecular mimicry are in theory too short to become recognized by cross-reactive antibodies or TCRs. These findings are in accordance with studies describing that up to 99.7% of heptapeptides from infectious agents can be identified in humans, which, however, does not result in the development of autoimmune diseases [261]. As no homology has been identified in the amino acid positions surrounding the cross-reactive regions, it remains to be determined whether the backbone of these sequences contributes, in theory, to generating a stable antibody/TCR-antigen interaction, which would favor a stable interaction. This remains to be elaborated.

Instead of focusing on the cross-reactive sequences between two components, it may be more relevant to examine the nature of TCRs/antibodies interacting with the antigenic binding sites [71,85,109,111,143,191,262,263,264,265]. As presented, when examining the contributions of epitopes for antibody binding, antibodies can be divided into groups: those that are very intolerant to amino acid substitutions in the epitope; those that have a high dependency of a single amino acid in combination with the epitope backbone; and those that tend to depend on a combination of specific contributions of epitope backbone and specific amino acid side chain contributions. Antibodies that are highly specific for the majority of amino acids presented in an epitope structure tend not to tolerate substitutions; based on this, it seems unlikely that specific antibodies intolerant of substitutions in the epitope structure may be cross-reactive and contribute to mediating the development of autoimmune diseases [71,109]. Instead, antibodies that mainly depend on epitope backbone, alternatively in combination with a few critical amino acid side chains in the epitope, appear most likely to be associated with cross-reactivity [110,117,120,262]. These assumptions are in accordance with studies describing RA-specific ACPA reactivity, as ACPAs, which recognize a specific amino acid in combination with epitope backbone, have been reported to be highly cross-reactive [110,120,262]. Based on this, and to determine true cross-reactivity between non-self-antigens and self-antigens, it would be of interest to thoroughly characterize the nature of assumed cross-reactive immune reactions and their role in autoimmunity, e.g., by a thorough analysis of the interaction of cross-reactive immune reactants in complex with their targets using high-resolution approaches.

In addition to epitope sequences and the nature of cross-reactive immune reactions, the role of the specific component may contribute to determining whether an antibody or TCR is associated with autoimmune diseases [54,55,56,57]. The findings of cross-reactive immune reactants do not necessarily indicate that these are associated with neither molecular mimicry nor disease pathogenesis in general [106,266,267,268]. This would require that cross-reactive immune reactants are pathogenic by nature, which has been widely discussed in the literature, as not all studies indicate that cross-reactive antibodies and autoantibodies in general, per definition, are pathogenic. In fact, several studies point to the fact that cross-reactive antibodies often function as disease markers rather than contribute actively to disease pathogenicity [106,266,267,268]. Moreover, examples have been reported where cross-reactivity provides protection for the host, referred to as heterologous immunity [269,270]. These observations are supported by findings describing that cross-reactive antibodies may even have a protective function in the development of various diseases [63,64,65,66,67]. Moreover, elderly humans or relatives of persons with autoimmune diseases often have an elevated incidence of autoantibodies, which are not associated with clinical autoimmune disease [271]. These findings are supported by the mere presence of self-reactive antibodies, also referred to as natural antibodies, which may be present in the sera of healthy individuals without causing tissue damage [271]. Finally, other factors associated with antibody characteristics, such as binding strength, antibody isotype, and glycosylation, may influence whether antibody self-reactivity is associated with autoimmunity [106].

Although several examples of homology between proteins and peptides from infectious agents and humans have been presented, the link between these agents and autoimmune diseases remains to be elaborated. For infectious agents to participate in the development of autoimmunity through molecular mimicry, four criteria must be addressed: homology between host proteins and infectious components, the presence of autoantibodies or autoreactive T or B cells, epidemiological data related to exposure to the infectious agent, and the development of autoimmunity and evidence in vivo in animal models [8]. Although it has been suggested, only a very limited number of examples of molecular mimicry address all criteria as the examples presented are often only qualitatively described, primarily focusing on cross-reactivity between proteins or the identification of cross-reactive regions identified between host components and infectious agents [7,8,42,88,186,189,190,204,206,209,230,240,251,253]. Hence, findings in the literature do not support the theory of molecular mimicry as being a major contributor to the development of autoimmunity, indicating that the precise role of molecular mimicry remains to be thoroughly investigated. Nevertheless, although the specific role between non-self and self-tissue cross reactivity remains to be elaborated, it cannot be neglected that viral and bacterial infections may play a central role in disease development. In particular, EBV appears to play a critical role in the development of several autoimmune diseases such as SLE, RA, and MS [17,28,34,35,36,160,163]. However, provided that molecular mimicry does not appear to contribute significantly to disease onset, other mechanisms must play central roles in the development of autoimmune diseases. Bystander activation and epitope spreading are mechanisms reported to be associated with autoimmunity, which are characterized by autoreactive T or B cells becoming activated without antigen recognition or though the complex formation of a self-antigen with a pathogen antigen, respectively, influencing the onset of autoimmune processes, where activation is a combination of an inflammatory environment, co-signaling ligands, and interactions with surrounding cells [31,272].

Collectively, these findings may support the proposed hypothesis that immune reactants to virus antigens may cross-react with human proteins but do not necessarily contribute to disease development.

## 7. Conclusions

As presented in this review and in the literature, infectious agents play an important role in the development of autoimmune diseases, leading to the generation of immune responses that may involve host proteins by one or more mechanisms. The examples presented here question whether cross-reactive antibodies are central for the development of autoimmune diseases through molecular mimicry or rather appear to be the result of cross-reactivity between antibodies generated to infectious antigens and host components. The current hypothesis describing that antibody self-reactivity does not necessarily cause and is not necessarily associated with autoimmunity through molecular mimicry is supported by thorough antibody characterization, illustrating that antibodies indeed have varying dependencies for reactivity. Collectively, the current findings do not support antibody-associated molecular mimicry as a main contributor in the development of autoimmune diseases. Based on this, the theory of molecular mimicry cannot stand alone; other mechanisms such as bystander activation, epitope spreading, and the exposure of cryptic antigens and superantigens appear to play a crucial role in the development of autoimmune diseases.

This review provides insights into the cross-reactive antibody responses between infectious non-self-antigens and self-tissue antigens, which are associated with autoimmune diseases. Based on this, more characterization studies are needed in order to understand the basic immunological mechanisms associated with molecular mimicry and the exact role cross-reactivity plays between non-self-infectious agents and self-antigens associated with autoimmunity, which also may contribute to understand the true nature of antibodies and the role they play in these diseases.

## Figures and Tables

**Figure 1 ijms-24-13609-f001:**
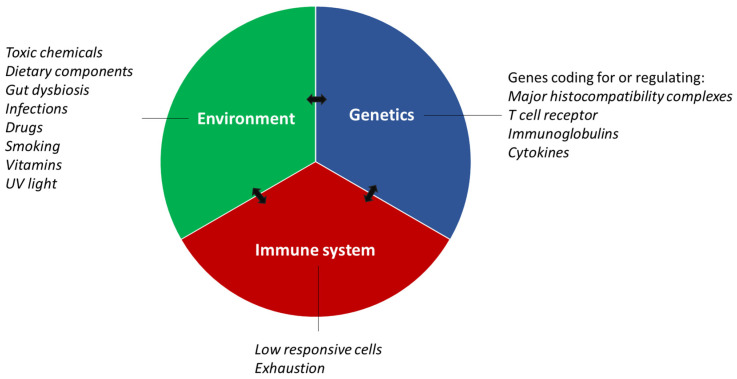
Factors triggering onset of autoimmune diseases. Genetic and environmental factors may along with acquired immune defects contribute to the onset of autoimmune diseases. Especially gene-environment interactions, influencing the immune system, significantly influence the risk of developing an autoimmune disease. UV, ultra violet.

**Figure 2 ijms-24-13609-f002:**
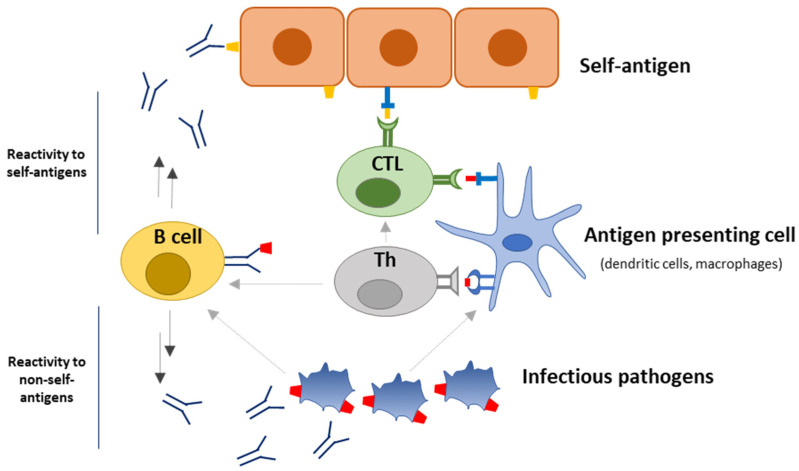
Schematic illustration of molecular mimicry associated with autoimmunity. Infectious pathogens express non-self-antigens (red box) that are similar in amino acid sequence or structure to self-antigens (yellow box). The immune response directed toward the infectious pathogens may cross-react with self-antigens, resulting in tissue damage, among others, by antibody effector functions stimulating complement activation along with antibody-dependent cytotoxicity, and cytotoxic T cells (CTL) inflicting cytotoxic tissue damage.

**Table 2 ijms-24-13609-t002:** Types of cross-reactive associated molecular mimicry shared between an infectious non-self-agent and its host. The % estimated structural identity and sequence identity is defined by mimic contact/natural ligand contact × 100%. In cases of structural and sequence identity between proteins, immune reaction is specific, whereas in cases representing with changes in the amino acid sequence, the immune reaction is cross-reactive.

	% Sequence Identity	% Structural Identity	Type of Reactivity	Reference
Epitope identity at the protein level between a host and an infectious agent, having “hijacked” a human protein	100	100	Specific	[8,90,91]
Similarity at the three-dimensional protein level between	2550	75–100	Cross-reactive	[8,41,91]
Similarity at the amino acid level in epitopes or critical hot spots (peptide level)	75–100	25–50	Cross-reactive	[8,84,85,88,91]
Structural “similarities” (mimotopes)	0–25	75–100	Cross-reactive	[8,89,91]

## Data Availability

Not applicable.

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
