# Peer review of "Antibody Cross-Reactivity in Auto-Immune Diseases"

_ijms, 2023, doi:10.3390/ijms241713609_

Round 1

Reviewer 1 Report

The abstract provides a good overview of the key points in the manuscript. However, it would be helpful to briefly mention the main ingredients or components examined in the study. This would help readers better understand the scope of the work.

The introduction contains several minor spelling, grammar, and formatting issues. It would be beneficial to ensure all key terms and author names are spelled out upon first use and presented in a consistent style throughout. The flow and organization of the background information could also be improved by grouping related ideas together into coherent paragraphs.

In some sections, information belonging to other sections is introduced. It would help clarity to ensure each section contains only relevant details and maintains a clear focus. Additionally, some statements lack supporting citations from the literature. Please review the referencing to ensure claims are properly substantiated.

The discussion could be strengthened by first presenting a clear summary of the key findings and their significance before relating them to prior studies. This would help highlight the original contributions of this work.

In the conclusion, consider adding a statement on the broader implications of the research and suggestions for potential next steps to build on these findings. This would provide helpful context for readers.

Finally, the manuscript would benefit from careful editing by a native English speaker to improve clarity and readability. There are some minor grammatical and word choice issues throughout that correction would help address.

Overall, this is an interesting study containing valuable insights. With some revisions to improve clarity, flow, and formatting, the quality of the work will be further enhanced. I hope these suggestions are helpful as you prepare the next version of the manuscript and look forward to seeing the revised paper.

There are some minor grammatical and word choice issues throughout that correction would help address.

Author Response

Dear editor and reviewers

Thank you for taking your time to read our manuscript draft and for your comments.

We realize that this review does not support the current hypothesis of molecular mimicry as a main contributor to the onset of autoimmunity. Thorough analysis of reported molecular mimicry examples reveals that many of these are sparsely described and does not take the nature of antibody cross-reactivity into account. Based on these observations, we attempted to evaluate data in a new light, focusing on immunological cross-reactivity between infectious non-self and self-tissue antigens. These observations provide interesting knowledge about cross-reactivity in various autoimmune diseases and may contribute to understand the true nature of antibodies and their role in these diseases.  

We have edited the manuscript according to the academic editors and the reviewers’ comments. Your thoughts and constructive input have significantly increased the quality of the manuscript.

Our comments to the individual reviewers and the academic editor are appended below.

Reviewer 1:

The abstract provides a good overview of the key points in the manuscript. However, it would be helpful to briefly mention the main ingredients or components examined in the study. This would help readers better understand the scope of the work.

Response: The abstract has been rephased and now mentions main components examined in this review.

The introduction contains several minor spelling, grammar, and formatting issues. It would be beneficial to ensure all key terms and author names are spelled out upon first use and presented in a consistent style throughout. The flow and organization of the background information could also be improved by grouping related ideas together into coherent paragraphs.

Response: Minor errors in the introduction have been corrected. The flow of the introduction has been altered and related subjects are grouped together in new paragraphs.

In some sections, information belonging to other sections is introduced. It would help clarity to ensure each section contains only relevant details and maintains a clear focus. Additionally, some statements lack supporting citations from the literature. Please review the referencing to ensure claims are properly substantiated.

 Response: Subjects related to each other have been grouped. Additional references have been added where needed. 

The discussion could be strengthened by first presenting a clear summary of the key findings and their significance before relating them to prior studies. This would help highlight the original contributions of this work.

Response: The discussion has been rewritten to focus even more on the cross-reactivity between non-self and self. Moreover, original contributions of this review have been clarified first in the discussion. 

In the conclusion, consider adding a statement on the broader implications of the research and suggestions for potential next steps to build on these findings. This would provide helpful context for readers.

Response: amended as requested. Additional thoughts about how to support these findings and proceed with these studies have been added to the conclusion.

Finally, the manuscript would benefit from careful editing by a native English speaker to improve clarity and readability. There are some minor grammatical and word choice issues throughout that correction would help address.

Response: The manuscript has been carefully read and grammatical errors and word choice issues have been corrected.  

Overall, this is an interesting study containing valuable insights. With some revisions to improve clarity, flow, and formatting, the quality of the work will be further enhanced. I hope these suggestions are helpful as you prepare the next version of the manuscript and look forward to seeing the revised paper.

Reviewer 2 Report

The authors review how structural similarities between infectious non-self and tissue antigens contribute to immunological autoreactivity and autoimmune diseases, giving examples and lists of concerned sequences/structures in specific diseases.

Major remark

Antibody self-reactivity does not necessarily cause and is not necessarily associated with autoimmunity. Other intrinsic factors, such as the binding strength, the isotype and glycosylation of antibody, and extrinsic factors, such as the duration of antibody response, may determine whether there is a relationship between the two. Self-reactive antibodies may even be protective against autoimmune events. Some thoughts of these aspects of antibody function would greatly improve the scientific value of the manuscript.

In Table 1. the mechanism description is too lengthy, too many “e.g.”-s; the authors should try to define the mechanisms more concisely. Conventionally, these mechanisms are defined following hypersensitivity reaction classifications, therefore it would make sense to either follow the convention or otherwise give reasons how the authors’ classification is better than the conventional.

Minor remarks

Line 373 [Chunga, Choi] is probably a reference not converted to numbers.

Line 499 Sentence is incomplete, sounds more like a title.

Author Response

Dear editor and reviewers

Thank you for taking your time to read our manuscript draft and for your comments.

We realize that this review does not support the current hypothesis of molecular mimicry as a main contributor to the onset of autoimmunity. Thorough analysis of reported molecular mimicry examples reveals that many of these are sparsely described and does not take the nature of antibody cross-reactivity into account. Based on these observations, we attempted to evaluate data in a new light, focusing on immunological cross-reactivity between infectious non-self and self-tissue antigens. These observations provide interesting knowledge about cross-reactivity in various autoimmune diseases and may contribute to understand the true nature of antibodies and their role in these diseases.  

We have edited the manuscript according to the academic editors and the reviewers’ comments. Your thoughts and constructive input have significantly increased the quality of the manuscript.

Our comments to the individual reviewers and the academic editor are appended below.

Reviewer 2:

The authors review how structural similarities between infectious non-self and tissue antigens contribute to immunological autoreactivity and autoimmune diseases, giving examples and lists of concerned sequences/structures in specific diseases.

Major remark

Antibody self-reactivity does not necessarily cause and is not necessarily associated with autoimmunity. Other intrinsic factors, such as the binding strength, the isotype and glycosylation of antibody, and extrinsic factors, such as the duration of antibody response, may determine whether there is a relationship between the two. Self-reactive antibodies may even be protective against autoimmune events. Some thoughts of these aspects of antibody function would greatly improve the scientific value of the manuscript.

Response: Thank you for your constructive comment. Your thoughts about antibody self-reactivity has been elaborated in the discussion in the context of antibody cross-reactivity.  

In Table 1. the mechanism description is too lengthy, too many “e.g.”-s; the authors should try to define the mechanisms more concisely. Conventionally, these mechanisms are defined following hypersensitivity reaction classifications, therefore it would make sense to either follow the convention or otherwise give reasons how the authors’ classification is better than the conventional.

Response: We acknowledge that table 1 was confusing and did not follow the conventional terms. To avoid confusion a new table 1 has been introduced primarily focusing on major mechanisms directly associated with autoimmunity, e.g. loss of tolerance, bystander activation, epitope spreading and potentially molecular mimicry as well.    

Minor remarks

Line 373 [Chunga, Choi] is probably a reference not converted to numbers.

Line 499 Sentence is incomplete, sounds more like a title.

Response: The references have been numbered, thank you for noticing. Line 499 was initially intended as a subheading for the discussion, but has been deleted to avoid confusion.